# Phytochemical Analysis of the Fruit Pulp Extracts from *Annona crassiflora* Mart. and Evaluation of Their Antioxidant and Antiproliferative Activities

**DOI:** 10.3390/foods11142079

**Published:** 2022-07-13

**Authors:** Natale Cristine C. Carvalho, Odair S. Monteiro, Claudia Q. da Rocha, Giovanna B. Longato, Robert E. Smith, Joyce Kelly R. da Silva, José Guilherme S. Maia

**Affiliations:** 1Programa de Pós-Graduação em Química, Universidade Federal do Pará, Belém 66075-110, Brazil; natalecrist@gmail.com; 2Programa de Pós-Graduação em Química, Universidade Federal do Maranhão, São Luís 65085-580, Brazil; odair.sm@ufma.br (O.S.M.); rocha.claudia@ufma.br (C.Q.d.R.); 3Programa de Pós-Graduação em Ciências da Saúde, Universidade São Francisco, Bragança Paulista 12916-900, Brazil; giovanna.longato@usf.edu.br; 4Department of Science, Park University, Parkville, MO 64152, USA; robert.smith@fda.hhs.gov

**Keywords:** marolo fruit, phenolic compounds, antioxidant and antiproliferative activities, volatile concentrate

## Abstract

*Annona crassiflora* Mart., the marolo fruit of the Cerrado biome, is one of the most frequently consumed species from the Brazilian Midwest. This study aimed to evaluate the chemical composition and the antioxidant and cytotoxic properties of the fruit pulp of *A. crassiflora* collected at Chapada das Mesas, Maranhão, Brazil. The volatile concentrate was identified as mainly ethyl octanoate, ethyl hexanoate, and methyl octanoate. From the ethanol (LFP-E) and ethyl acetate (LFP-A) extracts were identified phenolic acids (*p*-coumaric, gallic, quinic, and ferulic), flavones and derivatives (apigenin, epicatechin, 2′-5-dimethoxyflavone, 3′,7-dimethoxy-3-hydroxyflavone, kaempferol-3-O-glucoside and 3-O-rutinoside, quercetin-3-O-glucoside, procyanidin B2, and rutin), aporphine alkaloids (xylopine, stephagine, and romucosine), and acetogenin (annonacin). For the LFP-E and LFP-A extracts, the total phenolic compound values were 15.89 and 33.16 mg GAE/g, the flavonoid compound content values were 2.53 and 70.55 mg QE/g, the DPPH radical scavenging activity showed EC_50_ values of 182.54 and 57.80 µg/mL, and the ABTS radical activity showed TEAC values of 94.66 and 192.61 µM TE/g. The LFP-E extract showed significant cytotoxicity and cell selectivity for the U251-glioma strain, presenting a GI_50_ value of 21.34 µg/mL, which is close to doxorubicin (11.68 µg/mL), the standard chemotherapeutic drug. The marolo fruit seems to be a promising source for developing innovative and healthy products for the food industry.

## 1. Introduction

Annonaceae has been listed among the most diversified families due to its heterogeneity and abundance in the world’s tropical forests. It is a pantropical plant family of ca. 2450 species of trees and lianas, well represented in the neotropical flora with ca. 950 species. The Annonaceae comprise 26 genera and ca. 260 species in Brazil, known for their edible fruits and medicinal properties, where *Annona*, *Guatteria*, and *Xylopia* are the most common genera [1,2]. Some paper reviews have highlighted the traditional medicinal uses, phytochemical and pharmacological studies, and toxicity of most Annonaceae species, which indicate the presence of bioactive compounds exhibiting antimicrobial, insecticide, antiparasitic, pesticide, vermicide, and cytotoxic properties, among others [3,4]. Annonaceae are significant from an economic point of view given the numerous ways of using their species, such as fruits in food and cooking, rich in lipids and carbohydrates, in addition to diterpenes, acetogenins, alkaloids, and essential oils, among the constituents of secondary metabolism with a more outstanding contribution to the biological activities already observed, and which have served as primary compounds in the production of new drugs [5,6,7].

*Annona crassiflora* Mart. (syn. *A. macrocarpa* Barb. Rodr.), known as “marolo”, “araticum”, and “bruxo-da-quaresma”, is a medium-sized tree, 4 to 8 m tall, trunk usually 20 to 30 cm in diameter. The fruit is a subglobose berry with a 12–15 cm diameter and up to 2 kg in weight, oval to a rounded shape. It is one of Brazil’s twenty most frequently consumed fruits of the Cerrado biome, and its maturation occurs between February and April. When ripe, it has an aromatic and pleasant white-yellow pulp with numerous elliptical-shaped seeds [8]. The “marolo” fruits are highly appreciated by the native population and present unique sensory features, such as the intense flavor and exotic aroma, as well also used in the treatment of diarrhea, venereal and parasitic diseases, wounds, ulcers, cancer, and rheumatism. Its pulp is a rich source of dietary fiber, nutrients, and bioactive compounds, such as carotenoids, polyphenolics, tocopherols, flavonoids, and some vitamins and minerals [9,10,11,12]. These qualities make this fruit a promising source for developing innovative and healthy products for the food industry, and also, the fruit by-products and other plant parts are potential sources of value-added compounds [13,14,15,16]. *Annona crassiflora* fruit pulp showed 8.37 °Brix soluble solids, titratable acidity of 0.66 g of citric acid per 100 g. The amounts of metals, moisture, proteins, fiber, and sugars changed during the maturation of the fruit. The total amounts of sugars, as well as soluble and insoluble pectin and other parameters, were affected by the storage temperature of the fruits [17,18,19].

Annonacin and squamocin are acetogenins (ACGs) present in *A. crassiflora* and other Annonaceae species with neurotoxic action, which produce aggression to specific regions and cellular elements of the central nervous system. ACGs lead to some pathologies, such as an atypical Parkinson’s disease caused by the death of dopaminergic neurons, a decrease in energy production [20], and no response to the use of the L-DOPA standard medication [21]. The excessive consumption of “graviola” (*Annona muricata*) pulp is possibly related to the appearance of the etiology for some forms of an atypical Parkinson’s disease in Guadalupe, the French Caribbean [20,21]. There is insufficient data to determine the dose of ACGs or the fruits amount of Annonaceae that is neurotoxic. However, it seems unlikely that it is overconsumed in Brazil because the per-capita incidence of Parkinson’s disease (PD) in Brazil is less than half that of the USA, even though very little “graviola” or other Annonaceae are consumed there [22]. Squamocin and annonacin concentrations in another sample of lyophilized marolo fruit pulp and seeds were 0.403 and 142 mg/g of dry weight for the first and 0.334 and 5.905 mg/g of dry weight for the second, respectively [23]. The annonacin content determined in the fruit pulp of *A. crassiflora* was below that found for the fruit pulp of *A. muricata* L. (graviola), which was 0.768 mg/g of dry weight, but was above the value obtained for the fruit pulp of *A. squamosa* L. (atemoya), of 0.0038 mg/g of dry weight [24]. For the squamocin content, the value determined in *A. crassiflora* was above those found in the fruit pulp of *A. muricata* and *A. squamosa*, which were 0.0045 and 0.068 mg/g of dry weight, respectively [24].

Although there are many works on the characterization, application, and mechanisms of action in vitro and in vivo of the bioactive compounds from the leaves of *Annona* species, there are few studies related to the chemical composition and biological property of the extracts from *Annona* fruits. This work aimed to investigate the chemical composition and the potential antioxidant and antiproliferative effects exerted by the hydroethanolic extract of the *A. crassiflora* fruits.

## 2. Materials and Methods

### 2.1. Chemicals

The reagents, solvents, and standard compounds were purchased from the Sigma-Aldrich Chemical Company (St. Louis, MO, USA).

### 2.2. Plant Material and Extractions

Semi-ripe fruits of *Annona crassiflora* were collected at Parque Nacional Chapada das Mesas (7°20′16″ S/47°28′04″ W), Carolina, Maranhão, Brazil, March 2016, and transported to Laboratório de Engenharia de Produtos Naturais (LEPRON), Universidade Federal do Pará, Belém, Brazil. The leaves and flowers of *A. crassiflora* collected for taxonomic purposes, were identified and deposited in the Emílio Goeldi Herbarium, Belém Pará state, Brazil, under number MG 222438. The lyophilized fruit pulp (LFP, 50 g) was exhaustively extracted with *n*-hexane (500 mL), and then the LFP residue was extracted with a hydroethanolic solution (300 mL, ethanol 70%), thereby yielding the LFP-E extract. Subsequently, the LFP-E extract was fractionated with ethyl acetate (300 mL) to furnish the LFP-A extract. The LFP-E and LFP-A extracts were dried and used for the chemical and biological tests. The lyophilized fruit pulp (10 g) was also subjected to microdistillation-extraction [25] to obtain its volatile concentrate using *n*-pentane (99% HPLC grade, 3 mL) (Sigma-Aldrich, São Paulo, Brazil) as the solvent.

### 2.3. Chemical Characterization of A. crassiflora Extracts

A clean-up step was performed to remove contaminants for the HPLC-ESI-IT-MS/MS and FIA-ESI-IT-MSn analyses. The LFP-E (hydroethanolic) and LFP-A (ethyl acetate) extracts were purified by solid-phase extraction (SPE) using Phenomenex Strata C18 cartridges (500 mg, stationary phase) that were previously activated with MeOH (5 mL) and equilibrated with MeOH:H2O (5 mL,1:1, *v*/*v*). The purified extracts were eluted from cartridges using MeOH:H2O (5 mL, 1:1, *v*/*v*), filtered through a 0.22 μm PTFE filter, and dried. Then, the extracts were diluted to 10 μg/mL using the HPLC solvent. Aliquots of 20 μL were injected directly into the HPLC-ESI-IT-MS/MS and FIA-ESI-IT-MS^n^.

The LFP-E and LFP-A extract analyses were performed on an HPLC-ESI-IT-MS mass spectrometer LCQ Fleet Thermo Scientific using a Kinetex C18 (4.6 × 100 mm, 100 Å and 5μm) analytical column for the LC separation. The linear gradient elution from two mixtures was used in the mobile phase: 0.1% formic acid in water (A) and 0.1% formic acid in acetonitrile (B). A gradient elution starting from 10% to 100% of B for 10 min was used with a flow rate of 1.0 mL/min. The samples emerging from the HPLC system were analyzed online by ESI-MS in the negative ion mode in series with a UV detector. For the FIA-ESI-IT-MSn analysis, the direct flow infusion of the samples was performed on a Thermo Scientific LTQ XL linear ion trap analyzer equipped with an electrospray ionization (ESI) source (Thermo, San Jose, CA, USA). A stainless steel capillary tube was used at 280 °C, a spray voltage of 5.00 kV, a capillary voltage of 90 V, a tube lens of 100 V, and a flow rate of 5 μL/mL. A full scan analysis was recorded in the range from 100 to 1000 *m*/*z*.

Multiple-stage fragmentations (ESI-MSn) were performed using the collision-induced dissociation (CID) method against the helium for ion activation. The first event was a full-scan mass spectrum to acquire data for the ions in that *m*/*z* range. The second scan event was an MS/MS experiment performed using a data-dependent scan on the [M-H] molecules from the compounds of interest at a collision energy of 30% and an activation time of 30 ms. The product-ions were then submitted to further fragmentation in the same conditions until no more fragments were observed. The identification of the different compounds in the chromatographic profiles of the hydroalcoholic (LFP-E) and ethyl acetate (LFP-A) extracts was performed by comparing their retention times and spectra with literature data.

The volatile concentrate of the lyophilized fruit pulp of *A. crassiflora* was submitted to GC and GC-MS analysis. It was performed on a GCMS-QP2010 Ultra system (Shimadzu Corporation, Tokyo, Japan) equipped with an AOC-20i auto-injector and the GCMS-Solution software containing standards libraries [26,27]. A Rxi-5ms (30 m × 0.25 mm; 0.25 μm film thickness) silica capillary column (Restek Corporation, Bellefonte, PA, USA) was used. The conditions of analysis were as follows: Injector temperature: 250 °C; Oven temperature programming: 60–240 °C (3 °C min^−1^); Helium as the carrier gas, adjusted to a linear velocity of 36.5 cm s^−1^ (1.0 mL min^−1^); split mode injection (split ratio 1:20) of 1.0 µL of the *n*-pentane solution; electron ionization at 70 eV; and ionization source and transfer line temperatures of 200 and 250 °C, respectively. The mass spectra were obtained by automatic scanning every 0.3 s with mass fragments in the range of 35–400 *m*/*z*. The retention index was calculated for all volatile components using a homologous series of C_8_-C_40_
*n*-alkanes (Sigma-Aldrich, Milwaukee, WI, USA) according to the linear equation of van den Dool and Kratz (1963) [28]. Individual components were identified by comparing their retention indices and mass spectra (molecular mass and fragmentation pattern) with those existing in the GCMS-Solution system libraries [26,27]. The quantitative data regarding the volatile constituents were obtained using a GC2010 Series gas chromatograph that was operated under similar conditions to those of the GC-MS system. A flame ionization detector (GC-FID) was used to quantify the relative amounts of individual components by peak-area normalization. Chromatographic analyses were performed in duplicate.

### 2.4. Antioxidant Capacity of the Fruit Pulp of A. crassiflora

#### 2.4.1. Determination of Total Phenolics Content

The amount of total phenolics (TP) of the LFP-E and LFP-A extracts was determined according to the Folin-Ciocalteu colorimetric procedure [29,30]. A calibration curve with gallic acid at concentrations of 1, 2, 4, 6, 8, and 10 µg/mL was prepared. The acid gallic solutions’ aliquots and the samples (500 µL) were mixed with Folin-Ciocalteu reagent (250 µL, 1N) and sodium carbonate (1250 µL, 75.0 g/L). After 30 min of reaction, the absorbance was read at 760 nm in a spectrophotometer UV/Visible (Shimadzu, UV 1800, Shimadzu Corporation, Tokyo, Japan) at 25 °C and in a dark environment. The LFP-E and LFP-A extracts were solubilized in methanol at initial concentrations of 20,000 µg/mL (for the extract) and 10,000 µg/mL (for the fractions) to induce an absorbance between 0.3 and 0.7. The total phenolic content was expressed as gallic acid equivalents in mg per g of extract (mg GAE/g).

#### 2.4.2. Determination of Total Flavonoids Content

The amount of total flavonoids (TF) of the LFP-E and LFP-A extracts was determined according to the aluminium chloride colorimetric procedure [31]. A calibration curve with the quercetin standard at concentrations of 0.625, 1.25, 2.5, 5, 10, and 20 µg/mL was prepared. The aliquots of the quercetin solutions (1000 µL) were mixed with aluminum chloride (1000 µL, 2%). After 30 min of reaction, the absorbance was read in a UV/Visible spectrophotometer (Shimadzu, UV 1800, Shimadzu Corporation, Tokyo, Japan) at 420 nm in a dark environment, and at 25 °C. The LFP-E and LFP-A extracts were solubilized in methanol at an initial concentration of 6000 µg/mL. The total flavonoid content was expressed as quercetin equivalents in mg per g of extract.

#### 2.4.3. DPPH Radical Scavenging Assay

The LFP-E extract was evaluated by the DPPH radical-scavenging assay [30]. DPPH is a stable dark violet free radical with a maximum absorption of 517 nm and is reduced by antioxidants. A stock solution of 2,2-diphenyl-1-picrylhydrazyl (DPPH; 0.5 mM) was prepared in ethanol. The solution was diluted to approximately 60 µM and measured an initial absorbance of 0.62 ± 0.02 at 517 nm (Shimadzu, UV 1800, Shimadzu Corporation, Tokyo, Japan) at room temperature. The absorbance was measured at the start of the reaction, every 5 min during the first 30 min, and then at 30 min intervals until constant absorbance values were observed (plateau of reaction, 2 h). The extract (50 µL ethanol) was mixed to 1950 μL of methanolic DPPH solution (0.5 mM). The standard curves were prepared with concentrations of Trolox (6-hydroxy-2,5,7,8-tetramethylchroman-2-carboxylic acid) (Sigma-Aldrich, St. Louis, MO, USA) of 1, 2, 4, 6, 8, and 10 µg/mL. The results were expressed as a 50% inhibitory concentration (IC_50_) and milligrams of Trolox (mgTE/g) equivalents per gram of the extract.

#### 2.4.4. ABTS Radical Cation Assay

The LFP-E extract was evaluated by ABTS radical cation assay [30,32]. ABTS radical cation was obtained by mixing 2,2′-azino-bis(3-ethylbenzothiazoline-6-sulfonic acid) solution (7 mM, 5000 μL) with potassium persulfate solution (140 mM, 88 μL), followed by incubation for 16 h in the dark at room temperature. After, ABTS radical cation solution was diluted in ultrapure water until reaching the absorbance of 0.70 ± 0.02 at 734 nm (ABTS work solution). Aliquots (20 μL) of extract prepared in ultrapure water were added to ABTS work solution (2020 μL) and the absorbance was measured in a spectrophotometer (Shimadzu, UV 1800, Shimadzu Corporation, Tokyo, Japan). The results were expressed in Trolox Equivalent Antioxidant Capacity (TEAC, μMTE/g). The TEAC value was calculated by measuring the area under the curve and plotting the percentage inhibition of the absorbance as a function of time. The area under the curve was calculated for one sample dilution, which had a final percentage inhibition between 20% and 80%.

### 2.5. Antiproliferative Assay

The MTT 3-(4,5-dimethyl-2-thiazol)-2,5-diphenyl-2H-tetrazolium bromide colorimetric assay was used to measure cellular metabolic activity [33]. The antiproliferative activity of the LFP-E and LFP-A extracts were tested against six cancer cell lines: U251 (brain, gliobrastoma), MCF-7 (breast, adenocarcinoma), PC-3 (prostate, adenocarcinoma), OVCAR-3 (ovarian, adenocarcinoma), HT-29 (colorectal, adenocarcinoma), and HEP-G2 (liver, hepatocellular carcinoma), and one non-tumor cell line, HaCaT (skin, keratinocyte). The cells were seeded in 96-well plates and treated with extract concentrations of 1.3, 3.2, 6.5, 12.5, 25.0, 50.0, and 100.0 μg/mL, and then incubated for 48 h at 37 ° C with 5% CO_2_. MTT was dissolved in RPMI medium (0.25 mg/mL), added to the plate, and incubated at 37 ° C with 5% CO_2_. After 24 h, the plates were solubilized with DMSO solution (5%) and stirred for 15 min. The determination of cell proliferation was performed using a microplate reader (EpochBiotek) at 570 nm. The 50% growth inhibition (GI_50_), used for cytostatic samples, was calculated by non-linear regression using the software ORIGIN 8.0 (OriginLab Corporation). Doxorubicin was used as the positive control. MTT analyses were performed in triplicate.

### 2.6. Statistical Analysis

The samples were assayed in triplicate, and the results are shown as the mean ± standard deviation. An analysis of variance was conducted, and the differences between variables were tested for significance by a Tukey test and Student’s t-test. Differences at *p* < 0.05 were considered statistically significant. The IC_50′_s values were calculated by nonlinear regression using the GraphPad program (version 5.0, Intuitive Software for Science, San Diego, CA, USA).

## 3. Results and Discussion

### 3.1. Composition of Fruit Pulp Extracts of A. crassiflora

The hydroethanolic (LFP-E) extract from the hydrophilized fruit pulp of *A. crassiflora* yielded 15.6%. The fractionation of the LFP-E extract (4 g), with n-hexane and ethyl acetate furnished the LFP-H (2.9%) and LFP-A (1.5%) extracts, respectively. The chemical profiles of the LFP-E and LFP-A extracts were analyzed using the LC-ESI-IT-MS technique (*m*/*z* 100–1000 Da) and the fragmentation data recorded are shown in Table 1. The main secondary metabolites identified were phenolic acids; flavonoid glycosides derived from quercetin, kaempferol, rutin, aporphine alkaloids; and tetrahydrofuran-type acetogenin (see Table 1 and Figure 1).

Compounds 1 (*m*/*z* 163), 2 (*m*/*z* 169), and 4 (*m*/*z* 193) undergo decarboxylation (loss of CO_2_, 44 Da) and correspond to *p*-coumaric, gallic, and ferulic acids [34,35]. Compound 3 (*m*/*z* 191), characterized by quinic acid, produced ion-fragments in *m*/*z* 173 and *m*/*z* 111, which correspond to H_2_O molecule loss and the ring-opening by a retro-Diels-Alder (RDA) mechanism [36]. Compounds 5 (*m*/*z* 269), 13 (*m*/*z* 297), and 14 (*m*/*z* 281) refer to apigenin, 3′,7-dimethoxy-3-hydroxyflavone, and 2′,5-dimethoxyflavone, respectively, and an RDA mechanism primarily fragmented all these flavonoid compounds [37]. The fragmentation of compound 6 (*m*/*z* 609) produced two main ion-fragments, the ion *m*/*z* 447 due to the loss of the deoxyhexose unit and the ion *m*/*z* 301 derived from the loss of the hexose-deoxyhexose unit, which is characteristic of rutin, a biflavonoid derived from quercetin with a glycosidic bond in carbon 3 of the central pyran-ring [38]. Compound 8 (*m*/*z* 593) was identified as kaempferol-3-O-rutinoside, which produced compound 7 (*m*/*z* 447), the kaempferol-3-O-β-D-glucoside, by the loss of a glycoside unit. The ion-fragment *m*/*z* 285 arose from the loss of another glycoside unit [39]. The two first fragmentations occur by the loss of the glycosyl units, thereby leaving only the structure of kaempferol that undergoes cyclization by the RDA mechanism—generating the ion *m*/*z* 152. Compound 9 (*m*/*z* 463) produced the ion-fragment *m*/*z* 301, which is related to glycoside unit loss. It was identified as quercetin-3-glycoside (isoquercetin) [40,41]. The mass spectrum of compound 10 (*m*/*z* 577) showed the ion *m*/*z* 425, which is derived from the loss of 152 Da by a retro-Diels-Alder fragmentation (RDA), the ion *m*/*z* 407 of a subsequent neutral loss of a water molecule, and the ion *m*/*z* 289, which is attributed to the interflavanic link fragmentation. The comparison between the ion-fragments allowed a structural proposal for the dimer B2-type procyanidin [42,43].

Compound 11 showed a precursor ion in *m*/*z* 289, and the subsequent fragmentation pattern, which is related to the ions *m*/*z* 271, 245, and 205, was attributed to the epicatechin. The ion of greatest intensity (*m*/*z* 245) is compatible with the molecule’s enol unit loss [44]. Compound 12 (*m*/*z* 294), generated ion-fragments arising from losses of CH_3_O (*m*/*z* 264), NH (*m*/*z* 249), CH_2_O (*m*/*z* 219), and CO (*m*/*z* 191). The observed fragmentations are consistent for the aporphine alkaloid xylopine—with *m*/*z* 294. The initial loss of the mass fragments 15 and 31 Da is essential to identify whether the nitrogen of the amine group is linked to hydrogen or methyl, respectively [45]. Still, through the fragmentation analysis, it was possible to identify stephalagine (compound 15, *m*/*z* 308) and romucosin (compound 16, *m*/*z* 322), also aporphine alkaloids. The presence of such alkaloids, xylopine, stephalagine, and romucosin, was previously predicted in the extracts of *A. crassiflora* since they are found in several Annonaceae species and considered chemotaxonomic markers for the family [46]. The fragmentation analysis of compound 17 furnished the ions *m*/*z* 397, 379, 361, 343, 327, and 309, which are related to cleavages between the linkages C_19_–C_20_ and C_15_–C_16_, and a series of fragments with H_2_O loss, thereby indicating a tetrahydrofuran ring at position C_16_–C_19_. The structure was identified as acetogenin annonacin [47,48]. Acetogenins are natural compounds isolated from Annonaceae species and derived from long-chain fatty acids (C_35_–C_37_) via a polyketide route, which shows a γ-lactone-type terminal ring and a tetrahydrofuran-type (THF) ring along the chain, in association with oxygenated functional groups, such as hydroxyls, ketones, and epoxides [47,49].

Previously, polar components of hydroethanolic extract of fruit pulp of *A. crassiflora* were investigated by direct electrospray ionization mass spectrometry (ESI-MS) in the negative ion mode. Characteristic ESI mass spectra with various diagnostic ions were obtained from the extract. Fumaric acid (*m*/*z* 115), malic acid (*m*/*z* 133), and some hexoses (*m*/*z* 161, 179, 341, 503, 683) were identified [50]. Also, a more recent HPLC-MS analysis revealed 10 phenolic compounds (catechin, epicatechin, rutin, quercetin, and protocatechuic, gentisic, chlorogenic, caffeic, and ferulic acids) in the fruit pulp of *A. crassiflora* [51].

### 3.2. Composition of Volatile Concentrate of Fruit Pulp of A. crassiflora

Volatile constituents are responsible for the characteristic aroma and flavor of fruits and are present in a wide range of concentrations, which are represented by different chemical classes. The constituents of the volatile concentrate in the fruit pulp of *A. crassiflora* were analyzed by GC-FID and GC-MS and are listed in Table 2. The yield of volatile concentrate was 0.4%. In total, 25 constituents were identified comprising more than 94% of the volatile concentrate. The primary components were ethyl octanoate (34.8%), ethyl hexanoate (29.4%), and methyl octanoate (18.3%), which comprised 82.5% of the volatile concentrate.

The odor description of ethyl hexanoate (fruity, sweet), ethyl octanoate (fruity, floral), and methyl octanoate (fruity, green) point them to a significant contribution to the characteristics of the “marolo” fruit aroma, and correspond to about 90% of the total identified compounds. These results agree with previous reports, which found that esters were more abundant in the lyophilized and fresh *A. crassiflora* fruit, thereby justifying their characteristics as essential compounds derived from the plant lipid metabolism [12,52]. A similar result regarding the abundance of ester derivatives was obtained from the analysis of the volatile concentrate of *Caryocar brasiliense* Cambess.(pequi) (Caryocaraceae), another vital fruit of the Brazilian Cerrado, where ethyl hexanoate and ethyl octanoate are also the main volatile constituents [53].

### 3.3. Antioxidant Capacity of Fruit Pulp of A. crassiflora

#### 3.3.1. Total Phenolics Content

The Folin–Ciocalteau assay allowed the estimation of flavonoids, anthocyanins, and other phenolic compounds present in the LFP-E and LFP-A extracts. The total phenolic content (TPC) for the LFP-E and LFP-A extracts were 15.89 ± 0.11 mg GAE/g dw and 33.16 ± 1.1 mg GAE/g dw, respectively. According to Vasco and coworkers [54], fruits extract are classified into three categories based on their total phenolic content: low (<1 mg GAE/g), medium (1–5 mg GAE/g), and high (>5 mg GAE/g). Therefore, the LFP-E and LFP-A extracts can be considered excellent sources of phenolic compounds. The results for the phenolic compound content in the LFP-E and LFP-A extracts of *A. crassiflora* fruit pulp were lower and similar to fruits collected in Goiânia, Brazil, which displayed a content of 31.08 ± 1.23 mg GAE/g [55]. However, the samples displayed higher TPC levels compared to fruits collected in Brasília, Brazil (5.80 ± 1.43 mg GAE/g) and Minas Gerais, Brazil (2.11 ± 0.60 mg GAE/g) [56,57].

The variation in the levels of total phenolic compounds analyzed in a given sample compared to the results in the literature may be due to the geographic and environmental conditions of the region of origin, physiological and genetic factors of the plant, preparation and storage conditions, and extraction method [54].

#### 3.3.2. Total Flavonoids Content

The estimation of the flavonoid compounds present in the LFP-E and LFP-A extracts was quantified based on the standard curve prepared with the compound quercetin. The flavonoid content for the LFP-E and LFP-A extracts were 2.53 ± 0.51 mg QE/g and 70.55 ± 0.5 mg QE/g, respectively. The total flavonoid content of the LFP-E (2.53 mg QE/g) extract was much lower than that obtained in a previous study, which was collected in Minas Gerais and displayed amounts of 40.0 ± 0.15 mg QE/g [58]. On the other hand, the content observed in the LFP-A extract was much higher than samples previously (0.631 and ±0.04611.64 mg QE/g) [10,56]. According to the literature, flavonoids are the predominant class of phenolic compounds present in the fruit pulp of *A. crassiflora* [59,60]. These data are also related to the higher antioxidant activity evaluated in the reducing power tests of DPPH and ABTS reagents.

#### 3.3.3. Antioxidant Activity by the DPPH and TEAC/ABTS Assays

Phenolic compounds exhibit various functional properties due to their antioxidant capacity, acting as reducing agents, hydrogen donors, transition metal chelators, reactive oxygen and nitrogen species (ROS/RNS) quenchers, enzyme inhibitors involved in oxidative stress, and regulators/protectors of endogenous defense systems [51]. The DPPH and TEAC/ABTS assays have been used to evaluate the antioxidant activity of extracts of food matrices. DPPH is applied to evaluate hydrophilic and lipophilic compounds, while TEAC/ABTS is widely used to determine the antioxidant activity of hydrophilic compounds. Both methods are based on sample antioxidants’ ability to reduce the radicals by electron transferences and/or hydrogen atoms measured by absorption, which decrease at 517 nm (DPPH) and 734 nm (ABTS) [61].

The DPPH radical scavenging activity showed the EC_50_ values 182.54 ± 7.3 µg/mL and 57.80 ± 3.9 µg/mL of the LFP-E and LFP-A extracts, respectively. In the ABTS method, the LFP-E and LFP-A extracts showed significant antioxidant activities through the discoloring action of the cation radical ABTS, whose results expressed Trolox equivalent antioxidant capacity (TEAC) values that were 94.66 ± 1.9 µM TE/g and 192.61 ± 2.8 µM TE/g, respectively. The results showed a significant statistical difference in both methods according to Student’s t-test (*p* < 0.0001) (Figure 2).

The antioxidant activity of fruit extracts can be classified in three distinct categories based on DPPH/EC_50_ values: significant (DPPH/EC_50_ ≤ 100 μg/mL), medium (100 μg/mL < DPPH/EC_50_ ≤ 316 μg/mL), and weak (DPPH/EC_50_ > 316 μg/mL) [62]. Therefore, the LFP-E and LFP-A extracts have medium to good antioxidant properties, respectively. In comparison to previous studies, the alcoholic extracts from the fruit pulp of two other samples of *A. crassiflora* have shown EC_50_ values of 148.82 ± 0.98 µg/mL [55] and 93.76 µg/mL [51], exhibiting a lower antioxidant action than the LFP-A extract (57.80 ± 3.9 µg/mL).

Regarding the ABTS assay, the LFP-A extract presented higher and lower antioxidant activities than other extracts of *A. crassiflora*, whose values were 131.58 ± 19.61 µM TE/g [63] and 231.79 ± 8.65 µM TE/g [51], respectively. In general, the antioxidant activity of *A. crassiflora* fruit pulp, determined in this work by the DPPH and TEAC/ABTS methods, is lower when compared with the results obtained for the fruit peel and seed, where there is a more significant presence of phenolic compounds: DPPH (peel: 1065.0 ± 4.45 µM TE/g; seed: 917.0 ± 7.76 µM/TE/g) and TEAC/ABTS (peel: 2022.13 ± 0.98 uM TE/g; seed: 190.54 ± 7.54 µM TE/g) [64].

#### 3.3.4. Antiproliferative Assay

The evaluation of the antiproliferative activity of the LFP-E extract from the fruit pulp of *A. crassiflora* was carried out in six human tumor cell lines: U251 (brain), MCF-7 (breast), PC-3 (prostate), OVCAR- 3 (ovary), HT29 (colon), and HEP-G2 (liver), and one non-tumor cell line, HaCaT (skin) (Table 3, Figure 3).

The results presented high values for 50% growth inhibition (GI_50_) in most tumor cell lines, demonstrating the low cytostatic activity. Except for the glioblastoma strain (U251), in which the LFP-E extract showed a significant antiproliferative effect, presenting a GI_50_ value of 21.34 µg/mL, close to that obtained for doxorubicin (11.68 µg/mL), the chemotherapeutic drug widely used to treat different types of cancer, and, in this work, was used as the positive control. Furthermore, LFP-E was at least 21 times less toxic for the non-tumor cell line (HaCaT) than doxorubicin, with GI_50_ values of >100 µg/mL and 4.79 µg/mL, respectively. The tumor cell selectivity is an essential factor for possible drugs since they exhibit low selectivity against cancer cells and can cause toxicity to normal cells (side effects) [65]. The antiproliferative activity observed for the LFP-E extract of *A. crassiflora* can be explained by the presence of the acetogenins annonacin and squamocin, which are compounds capable of preventing electron transport in the mitochondrial complex I through the inhibition of the NADH coenzyme, which is responsible for the production of cellular energy [23]. The LFP-A extract of *A. crassiflora* showed no antiproliferative effect.

Other hydroethanolic extracts of *A. crassiflora* leaves in a human tumor cell lineage were previously reported, with significant antiproliferative activity and GI_50_ values less than 10 μg/mL for various cell lines [66,67]. Interestingly, the peels and seeds of the *A. crassiflora* fruit contained similar phytochemicals [64]. Also, the extracts were found to have in vitro anticancer activities against multidrug-resistant ovary adenocarcinoma, glioma, breast, non-small cell lung cancer, prostate, ovary, colon, and leukemia.

## 4. Conclusions

The present study revealed that the extracts of the fruit pulp of *A. crassiflora* showed significant phytochemical constituents belonging to phenolic acids, flavonoids, aporphine alkaloids, acetogenins, and fatty acid esters classes. Many of the identified compounds are correlated to the observed antioxidant and antiproliferative activities, suggesting that the extracts could be used in the future for potential applications in functional foods, oxidative stress control, and cancer treatment. Also, the odors of ethyl hexanoate (fruity, sweet), ethyl octanoate (fruity, floral), and methyl octanoate (fruity, green) contribute significantly to the characteristics of the “marolo” fruit aroma.

## Figures and Tables

**Figure 1 foods-11-02079-f001:**
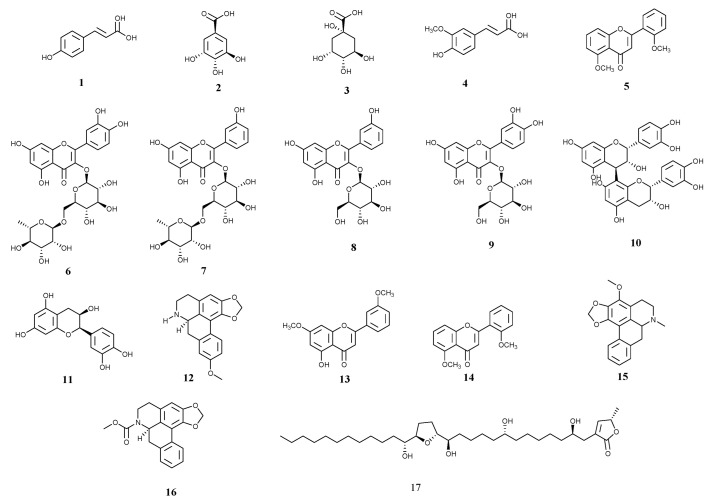
Constituents identified in the hydroethanolic and ethyl acetate extracts from fruit pulp of *A. crassiflora* (see Table 1).

**Figure 2 foods-11-02079-f002:**
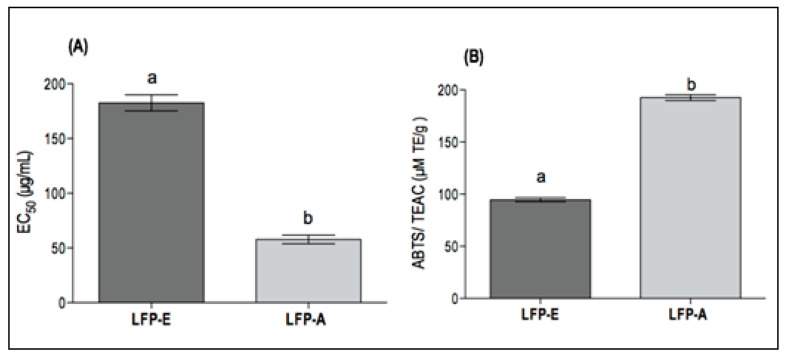
Antioxidant capacity of *A. crassiflora* extracts: (**A**) EC_50_ values of DPPH radical scavenging, (**B**) Total antioxidant capacity based on ABTS method. ^a,b^ Different letters represent statistical difference by Tukey test (*p* < 0.05).

**Figure 3 foods-11-02079-f003:**
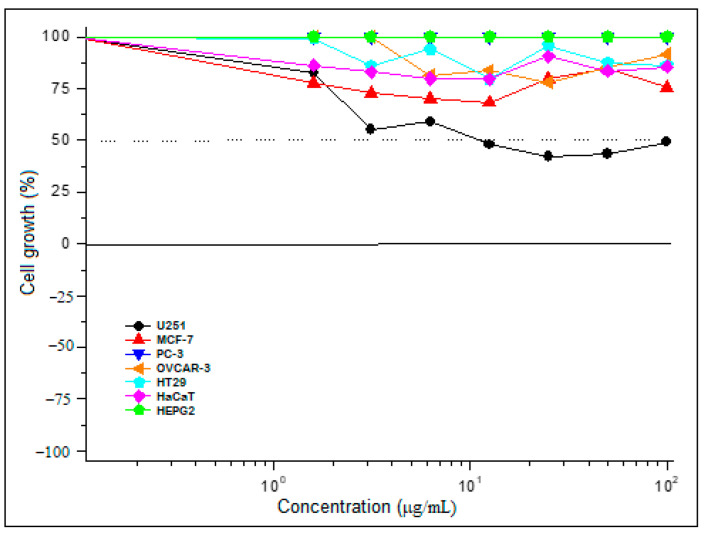
Growth inhibition of the tumor cell lines after LFP-E extract treatment (48 h).

**Table 1 foods-11-02079-t001:** Mass spectral characteristics of secondary compounds detected by HPLC-ESI-IT-MS in the *A. crassiflora* pulp fruit extracts at negative ionization mode *.

LFP-E and LFP-A Extracts	Compounds	RT (min)	[M-H]^-^ (*m*/*z*)	MS^n^ Fragments (*m*/*z*) ^c^
1	*p*-Coumaric acid ^a^	2.1	163	119(45), 93(100)
2	Gallic acid ^a^	2.1	169	125(25)
3	Quinic acid ^a,b^	2.3	191	173(40), 85(100)
4	Ferulic acid ^b^	2.4	193	178(35), 149(100), 134(25)
5	Apigenin ^b^	7.2	269	151(100), 117(35)
6	Rutin ^a^	12.3	609	463(25), 301(100)
7	Kaempferol 3-O-β-D-glucoside ^a,b^	12.9	447	285(100), 152(25)
8	Kaempferol-3-O-rutinoside ^a^	13.1	593	285(100), 255(45)
9	Quercetin-3-O-β-D-glucoside ^a^	14.2	463	301(100)
10	Procyanidin B2 ^a^	16.1	577	541(35), 425(35), 407(25), 289(100)
11	(-)-Epicatechin ^a^	18.3	289	271(100), 163(50)
12	Xylopine ^a,b^	18.6	294	264(100), 249(35), 219(45), 191(25)
13	3′,7-Dimethoxy-3-hydroxyflavone ^a,b^	19.6	297	265(100), 249(15), 183(18)
14	2′,5-Dimethoxyflavone ^a^	19.7	281	151(100)
15	Stephalagine ^a^	19.7	308	278(100)
16	Romucosine ^a^	19.9	322	267(100), 252(35)
17	Annonacin ^a^	21.2	595	471(100), 379(35), 361(28), 343(25)

* Identification by comparison of retention times and mass spectra data with reference compounds; ^a^ Identified in LFP-E extract, (hydroethanolic); ^b^ Identified in LFP-A extract (ethyl acetate); ^c^ Relative ionic abundance for each ion is in parentheses.

**Table 2 foods-11-02079-t002:** Constituents of volatile concentrate of lyophilized fruit pulp of *Annona crassiflora*.

Constituents	RI_C_	RI_L_	Concentrate (%)	Constituents	RI_C_	RI_L_	Concentrate (%)
*n*-Octane	799	800 ^a^	0.1	**Ethyl octanoate**	1195	1196 ^a^	**34.8**
*n*-Hexanol	861	863 ^a^	0.2	Ethyl (*2E*)-octenoate	1245	1245 ^a^	0.3
2-Heptanone	888	889 ^a^	0.1	Isopentyl hexanoate	1249	1252 ^b^	0.8
Methyl hexanoate	920	921 ^a^	2.4	(*2E*)-Decenal	1260	1260 ^a^	2.0
Hexanoic acid	965	967 ^a^	0.1	(*2E,4Z*)-Decadienal	1290	1292 ^a^	0.5
**Ethyl hexanoate**	996	997 ^a^	**29.4**	Methyl decanoate	1321	1323 ^a^	0.1
*n*-Octanol	1061	1063 ^a^	0.1	Hexenyl (*3Z*)-hexanoate	1381	1382 ^b^	0.2
Propyl hexanoate	1094	1096 ^b^	0.1	Hexyl hexanoate	1386	1390 ^b^	0.1
*n*-Nonanal	1100	1100 ^a^	0.1	Ethyl decanoate	1395	1395 ^a^	0.7
**Methyl octanoate**	1122	1123 ^a^	**18.3**	Ethyl dodecanoate	1596	1598 ^b^	0.5
Isobutyl hexanoate	1149	1149 ^a^	0.1	Ethyl tetradecanoate	1794	1795 ^a^	1.8
Octanoic acid	1164	1167 ^a^	1.1	Ethyl hexadecanoate	1991	1992 ^a^	0.1
Butyl hexanoate	1185	1186 ^a^	0.2				
Total (%)							94.2

RI_C_ = Calculated Retention Index (Rxi-5ms column); RI_L_ = Literature Retention Index; Bold = Main constituents; ^a^ Adams 2007 [26]; ^b^ Mondello 2011 [27].

**Table 3 foods-11-02079-t003:** Antiproliferative activity of *A. crassiflora* hydroethanolic extract against human tumor cell lines.

Extract/Standard	Cell Lines (GI_50_ µg/mL)						
	U251	MCF-7	PC-3	OVCAR-3	HT29	HaCaT	HEPG2
LFP-E	21.34	>100	>100	>100	>100	>100	>100
Doxorubicin	11.68	3.09	24.46	53.92	26.16	4.79	27.53

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
