# Peer review of "Phytochemical Analysis of the Fruit Pulp Extracts from Annona crassiflora Mart. and Evaluation of Their Antioxidant and Antiproliferative Activities"

_foods, 2022, doi:10.3390/foods11142079_

Round 1
Reviewer 1 Report
Manuscript is well and clearly written and the results are clearly presented. The data are new and have significance for investigation of food.
I have only few suggestions for Authors:
- Chromatogram of extract should be added (e.g. to Supplementary material) to support the results
- Authors investigated two type of extracts; however only one of them was used for antiproliferative assay. Short explanation why only LFP-E extract was taken for the tests should be added to the manuscript
- 3. Y-axis should be in narrower range (e.g. from 0 to 100%). Moreover, error bars should be added.
Author Response
Reviewers’ comments
Question 1. Chromatogram of extract should be added (e.g. to Supplementary material) to support the results.
Response: We added chromatogram (Supplementary Material)
Question 2. Short explanation why only LFP-E extract was taken for the tests should be added to the manuscript.
Response: The reviewer has been answered. See lines 455 to 459 of the revised manuscript.
Question 3. Y-axis should be in narrower range (e.g. from 0 to 100%). Moreover, error bars should be added.
Response: We appreciate the reviewer's comment, but this form of presentation of the antiproliferative trial is the most common in the literature.
Example: https://doi.org/10.21577/0103-5053.20210036.

Reviewer 2 Report
Manuscrisul cutitlul “Phytochemical analysis of the fruit pulp extracts from Annona crassiflora Mart. and evaluation of their antioxidant and antiproliferative activities ”is one that characterizes the pulp of Annona crassiflora Mart fruits. from the phytochemical point of view, the antioxidant and antiproliferative activity.
From my point of view, this manuscript would be more suitable for the Journal of Horticulture or Agronomy.
The abstract briefly describes the results obtained. The conclusion from the abstract is far too general given the context of the manuscript. I would recommend introducing the purpose, clearly described, in the abstract.
The samples were collected in 2016, then they were lyophilized. When were the tests done?
I would recommend a picture with the samples to be analyzed, if any.
I recommend a table with the results presented in section 3.3
Why were the fruits harvested half-ripe?
The results are well presented, but the discussions are very poor. They could be improved.
Very short and general conclusion
Author Response
Reviewers’ comments
Question 1. The abstract briefly describes the results obtained. The conclusion from the abstract is far too general given the context of the manuscript. I would recommend introducing the purpose, clearly described, in the abstract.
Response: Our apologies to Reviewer 2 for not understanding the recommendation for this item. However, the Abstract is longer than 200 words, and the Conclusions have been extended.
Question 2. The samples were collected in 2016, then they were lyophilized. When were the tests done?
Response: The results are from the NCCC doctoral thesis, whose experimental work was conducted from 2016 to 2019. However, we are only now submitting it for publication.
Question 3. I would recommend a picture with the samples to be analyzed, if any.
Response: We will include a photo of the fruit and the isolated constituents in the Graphical Abstract.
Question 4. I recommend a table with the results presented in section 3.3.
Response: The data discussion and figure 2 seems to us sufficient to express the fruit's antioxidant capacity. So that the manuscript doesn't look too repetitive.
Question 5. Why were the fruits harvested half-ripe?
Response: When ripening, the fruits fall to the ground and are eaten by rodents. We had to collect them semi-ripe and wait a few days to ripen fully.
Question 6. The results are well presented, but the discussions are very poor. They could be improved.
Response: We do not highlight the Results from the Discussion. Both are together and satisfactorily complemented with data insertion from many other related works.
Question 7. Very short and general conclusion.
Response: It was extended.

Reviewer 3 Report
The authors present the research work entitled: Phytochemical analysis of the fruit pulp extracts from Annona crassiflora Mart. and evaluation of their antioxidant and antiproliferative activities. This study is interesting and of importance for food and pharmaceutical sector. However, this paper, must be improved before further consideration.
1. In line 23 the metabolite squamocin is mentioned and does not appear in table 1, where the composition of the fruit pulp is shown.
2. Lines 64-68 show unnecessary information, since in lines 59-61 the composition of the pulp is mentioned. In addition, the manuscript does not discuss the storage of the fruit.
3. The paragraph (line 69) must be written in such a way that it connects with the information that precedes it for a better understanding. (“Introduction section”: must be restructured and improved).
4. Line 99 “Plant material and extractions” section.
The procedure is not well understood (line 105-112).
How was the n-hexane removed from the extract?
What do you mean by the residue or was the extract centrifuged?
How do you ensure that there are no remnants of the solvents used for the extractions?
5. Line 187 “DPPH radical scavenging assay” section
The extract, in which solvent was it dissolved?
6. Lines (275-278) delete information
7. Line 239 “Composition of fruit pulp extracts of A. crassiflora” section
All these metabolites have been identified in other species of Annonaceae?
What is the difference of metabolites with other species of Annonaceae? It would be interesting to add this information.
8. The results of lines 354, 355 and 370-374 are consistent with what is obtained in table 1.
9. Why there are differences in the results of antioxidant activity, quantification of phenols and flovonoids in both extracts? The extract (LFP-A) shows better results. It would be interesting to discuss this point.
10. Line 458. So, the leaves are better antiproliferative than the fruit.
The leaves do not have acetogenins?
11. Lines 461-464. What are these results due to?
I think the article is fine. However, in the discussion of the results, the comparison with other species of Annonaceae is missing. In addition, it would be interesting to quantify the identified acetogenins and compare the result with other species or similar fruits and thus be able to find out if the low antiproliferative activity can really be attributed completely to the presence and quantity of acetogenins.

Author Response
Reviewers’ comments
Question 1. In line 23 the metabolite squamocin is mentioned and does not appear in table 1, where the composition of the fruit pulp is shown.
Response: Abstract has been revised and lines 23 and 80-82 has been fixed. In fact, squamocin was not detected in the LFP-E and LFP-A extracts of this sample of A. crassiflora. Both annonacin and squamocin acetogenins were identified in the methanolic extract of another A. crassiflora sample previously published by us (see Reference 23, Tran et al., 2021, https://doi.org/10.1007/s12011-020-02320- 7). We thank the reviewer for this observation.
Question 2. Lines 64-68 show unnecessary information, since in lines 59-61 the composition of the pulp is mentioned. In addition, the manuscript does not discuss the storage of the fruit.
Response: Information in lines 59-61 (references 9-12) and 64-68 (references 17-19) regarding the pulp composition of the A. crassiflora fruit are complementary and not repetitive. Fruit storage was mentioned in a referenced article (see ref. 18) and it was not a subject of this paper.
Question 3. The paragraph (line 69) must be written in such a way that it connects with the information that precedes it for a better understanding. (“Introduction section”: must be restructured and improved).
Response: The presence of acetogenins in Annonaceae is a matter to be highlighted, mainly because they are present in the fruit of A. crassiflora, and there are reports of their neurotoxic action by the abundant consumption of A. muricata fruits elsewhere. In our understanding, the paragraph relating to lines 69-82 is wholly connected to the previous paragraph, lines 50-68, which highlights the nutrients and bioactive compounds previously identified in A. crassiflora.
Question 4.1. Line 99 “Plant material and extractions” section. The procedure is not well understood (line 105-112). How was the n-hexane removed from the extract? What do you mean by the residue or was the extract centrifuged?
Response: The reviewer interpreted the extraction process mentioned in lines 105-112 differently. The mentioned residue is from the lyophilized pulp of the fruit and not from the extract obtained with n-hexane. The procedure with n-hexane was done exclusively to remove the fatty material from the medium. We improved the text.
Question 4.2. How do you ensure that there are no remnants of the solvents used for the extractions? Response: Extracts obtained were kept in a desiccator/vacuum for three days before processing.
Question 5. Line 187 “DPPH radical scavenging assay” section. The extract, in which solvent was it dissolved?
Response: Ethanol. The text has been improved.
Question 6. Lines (275-278) delete information.
Response: The reviewer has been attended. We appreciate the correction.
Question 7. Line 239 “Composition of fruit pulp extracts of A. crassiflora” section. All these metabolites have been identified in other species of Annonaceae? What is the difference of metabolites with other species of Annonaceae? It would be interesting to add this information.
Response: Compounds identified in the LFP-E and LFP-A extracts are described in lines 254-322 and were made by comparison with spectral data obtained from other plant species. The description is supported by 17 references (35-51) identifying the plant species where these compounds were previously detected, including the Annonaceae family. We understand that it would be unnecessary to repeat this information.
Question 8. The results of lines 354, 355 and 370-374 are consistent with what is obtained in table 1.
Response: Yes, qualification (Table 1) and quantification (phenolic and flavonoid contents) by different and complementary analyses.
Question 9. Why there are differences in the results of antioxidant activity, quantification of phenols and flovonoids in both extracts? The extract (LFP-A) shows better results. It would be interesting to discuss this point.
Response: The variation in the levels of total phenolic compounds analyzed in a given sample compared to results in the literature may be due to the geographic and environmental conditions of the region of origin, physiological and genetic factors of the plant, preparation and storage conditions, and the extraction method (solvente polarity, for example). See reference 54.
Question 10.1 Line 458. So, the leaves are better antiproliferative than the fruit.
Response: Yes, higher content of phenolic compounds.
Question 10.2. The leaves do not have acetogenins?
Response: No, in Annonaceae species.
Question 11. Lines 461-464. What are these results due to?
Response: Higher content of phenolic compounds.

Round 2
Reviewer 2 Report
The manuscript can be published in its current form